# Coherent quantum control of nitrogen-vacancy center spins near 1000 kelvin

Gang-Qin Liu[1], Xi Feng[1], Ning Wang[1], Quan Li[1,2] & Ren-Bao Liu [1,2]

Quantum coherence control usually requires low temperature environments. Even for nitrogen-vacancy center spins in diamond, a remarkable exception, the coherence signal is limited to about 700 K due to the quench of the spin-dependent fluorescence at a higher temperature. Here we overcome this limit and demonstrate quantum coherence control of the electron spins of nitrogen-vacancy centers in nanodiamonds at temperatures near 1000 K. The scheme is based on initialization and readout of the spins at room temperature and control at high temperature, which is enabled by pulse laser heating and rapid diffusion cooling of nanodiamonds on amorphous carbon films. Using the diamond magnetometry based on optically detected magnetic resonance up to 800 K, we observe the magnetic phase transition of a single nickel nanoparticle at about 615 K. This work enables nano-thermometry and nano-magnetometry in the high-temperature regime.

[1] Department of Physics, The Chinese University of Hong Kong, Shatin, New Territories, Hong Kong, China. [2] Hong Kong Institute of Quantum Information Science and Technology, The Chinese University of Hong Kong, Shatin, New Territories, Hong Kong, China. These authors contributed equally: Gang-Qin Liu, Xi Feng. Correspondence and requests for materials should be addressed to Q.L. (email: liquan@phy.cuhk.edu.hk) or to R.-B.L. (email: rbliu@cuhk.edu.hk)

Quantum phenomena usually occur at low temperature. Quantum coherence control at high temperature is not only of fundamental interest by extending the boundary of the quantum regime but also of practical applications by enabling quantum sensing[1] in many important fields of nanoscience and nanotechnology. The high temperature sensing would be particularly relevant to, for example, thermoremanent magnetization of nanoparticles[2], magnetic records in petrology and planetary science[3], thermoelectric effects in nanostructures[4–7], heat-assisted magnetic recording[8,9], and thermo-plasmonics of nanoparticles[10–12], where the relevant temperature is often >500 K, with few nano-sensors available. Remarkably, the nitrogen-vacancy (NV) center in diamond has been demonstrated to have robust quantum coherence at room temperature[13,14] and even up to 700 K[15], which has stimulated many studies for quantum information processing[14] and quantum sensing[1,16].

The ground state of NV centers is a spin triplet, with a zero-field splitting (ZFS) $D \approx 2.87$ GHz at room temperature between the $|0\rangle$ and $|\pm 1\rangle$ states quantized along the NV axis (Fig. 1a). The ZFS depends on temperature $T$ with $dD/dT = -74$ kHz K$^{-1}$ near room temperature[17], which can be used to calibrate the temperature[18–21]. The NV center can be excited by a laser to a triplet state with $m_s$ conserved and then return to the ground state either by spin-conserving photon emission or by non-radiative relaxation (via intermediate singlet states by inter-system crossover). The non-radiative process is more efficient if the NV is in the $|\pm 1\rangle$ states and the intermediate singlet states have higher probability to relax to the $|0\rangle$ state, so a laser pulse can initialize NV centers to the $|0\rangle$ state in a few microseconds and the $|0\rangle$ state has stronger photoluminescence than the $|\pm 1\rangle$ states. After the initialization, microwave (MW) pulses can be applied to coherently manipulate the spin state, and then the spin state can be read out via spin-dependent fluorescence. The photon counts depend on the resonance between the MW pulses and the spin transitions, hence the optically detected magnetic resonance (ODMR).

However, it was found that both the overall photoluminescence intensity and the fluorescence contrast between different spin states dropped dramatically as the temperature increased to above 550 K and hence the ODMR signal of NV centers became invisible at above 700 K[15]. The loss of spin resonance signals at high temperature was ascribed to the enhancement of non-radiative relaxation of optically excited NV center state with $m_s = 0$, which prevented the spin readout at high temperature[15]. Nonetheless, the spin coherence is expected to be robust against the temperature increase[15,22].

Here we demonstrate quantum coherence control of the electron spins of NV centers in nanodiamonds (NDs) at temperatures near 1000 K. We overcome the problem of spin-dependent fluorescence quench by initializing and reading out the spins at room temperature and controlling them at high temperature, making use of pulsed laser heating and rapid diffusion cooling of

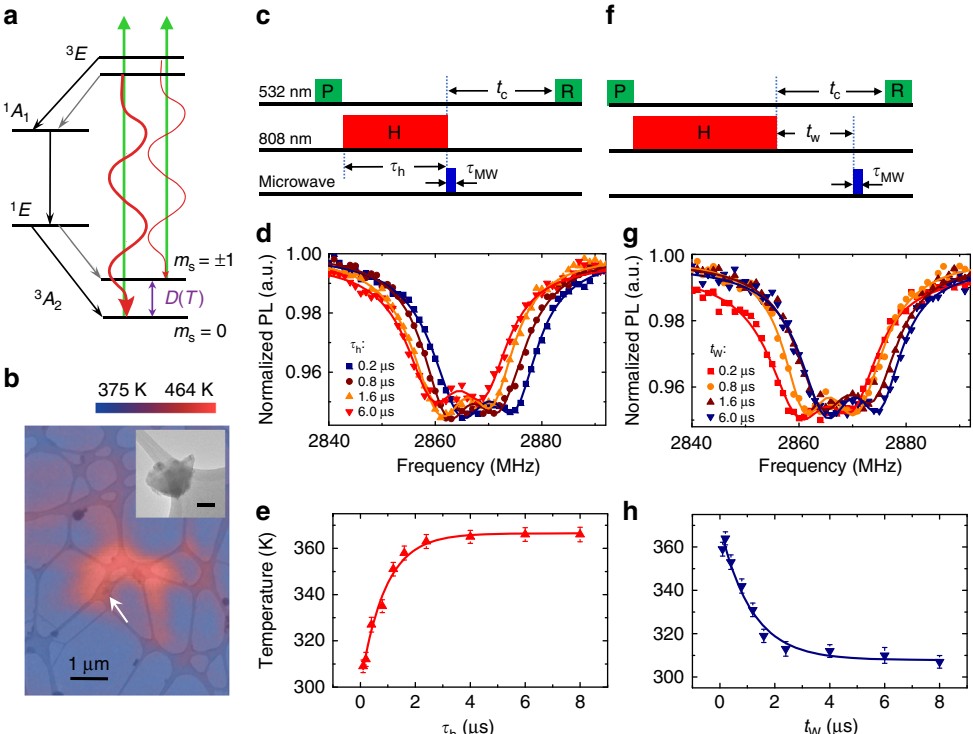

**Fig. 1** Fast temperature control of a nanodiamond (ND) by pulse NIR laser heating. **a** The energy level diagram of a nitrogen-vacancy (NV) center illustrating the spin polarization and readout. **b** Transmission electron microscopy (TEM) image of amorphous carbon film with NDs. The color contour shows the temperature of an ND (indicated by the arrow) as a function of the position of the NIR laser focus spot. The temperature is extracted from the zero-field splitting (ZFS) $D$ of the NV centers in the ND using the known $D$-$T$ relation in ref. [15]. Inset: a typical TEM image of an ND on a carbon film (scale bar: 50 nm). **c** Pulse sequence for characterizing the heating dynamics, including the 532 nm laser pulses for spin polarization (P) and readout (R), the 808 nm laser pulse for heating (H), and the microwave pulse for spin control. **d** Optically detected magnetic resonance (ODMR) spectra of an ND for various heating durations (for the pulse sequence in **c**). The cooling time $t_c = 10$ μs is fixed. **e** The ND temperature as a function of the heating pulse duration $\tau_h$, extracted from the ZFS $D$ obtained from the ODMR spectra in (**d**) using the $D$-$T$ relation in ref. [15]. **f** Pulse sequence for characterizing the cooling dynamics. The symbols are the same as in (**c**). **g** ODMR spectra of an ND for various waiting durations $t_w$, with a fixed heating duration $\tau_h = 10$ μs, and a fixed cooling time $t_c = 10$ μs (for the pulse sequence in **f**). **h** The cooling dynamics: the temperature of the ND as a function of the waiting period $t_w$. Error bars in (**e**) and (**h**) corresponding to standard fitting errors

NDs on amorphous carbon films. With an ND sensor working at high temperature now available, we carry out nano-magnetometry up to 800 K and observe the magnetic phase transition of a single nickel nanoparticle at about 615 K. This work, by integrating high temperature quantum control and nanoscale resolution, offers an approach to nano-thermometry and nano-magnetometry in the high temperature regime[2–12].

## Results

**High temperature ODMR (HiT-ODMR).** We measure ODMR[13] of NDs deposited on the amorphous carbon films on transmission electron microscopy (TEM) copper grids (Fig. 1b). To overcome the high temperature limit on spin initialization and readout, we initialize and read out the spins at low temperature (<500 K) and carry out the spin control at high temperature.

Rapid heating and cooling are required for such a scheme. The rapid heating is realized by the absorption of a near-infrared (NIR) laser pulse by the carbon film. The NIR laser pulse heats the carbon film near the focus spot and in turn heats the ND in the spot (see the color map in Fig. 1b). Since the heating spot is small, the carbon film and the ND rapidly cool down by heat diffusion through the carbon film when the heating laser is switched off. The 808 nm wavelength for the heating laser is chosen to minimize disturbance of the spin states and the charge states of the NV centers[23–27], so that the NIR laser can be simultaneously applied to maintain the temperature while the spins are manipulated by MW pulses. The highest temperature that is reached in each heating cycle can be tuned by adjusting the laser power and the heating pulse duration. To suppress the oxidation of the carbon film (which reduces the heating efficiency) during the laser heating, the samples, after the TEM imaging is obtained, are sealed in an argon (Ar) gas chamber (see Methods and Supplementary Note 2).

We first characterize the heating and cooling dynamics. Figure 1c shows the pulse sequence of characterizing the heating dynamics. A 532 nm laser pulse of 3-µs duration (P) is first applied to initialize the NV center spins in the ND at room temperature. Then, an NIR heating pulse (H) is applied for a duration $\tau_h$. A short MW pulse is applied immediately after the NIR pulse to flip the NV center spins; the MW pulse duration, $\tau_{MW} \sim$ tens of ns, is chosen to be much shorter than the heating and cooling times so that the temperature variation during the MW pulse is negligible. After a cooling time $t_c$, a 532 nm laser pulse (R) is applied to read out the fluorescence. The fluorescence depends on the spin state and hence on the resonance between the MW pulse and the NV center spin transitions. Typical ODMR spectra are shown in Fig. 1d for various heating pulse durations. The temperature of the ND is extracted from the ZFS shift of the spectra (obtained by double-Lorentzian-peak fitting). As summarized in Fig. 1e, the temperature as a function of the heating pulse duration ($\tau_h$) is well fitted by a rate equation with a heating rate (which depends on the laser power) and a cooling rate (due to heat diffusion through the carbon film) (see Methods and Supplementary Note 4). The rising time (about 2 µs) of the temperature is determined by the heat diffusion rate of the carbon film) and the saturated temperature is determined by the ratio of the heating rate to the diffusion rate. The cooling dynamics is studied similarly, but with a fixed heating pulse duration and a varying waiting time $t_w$ between the heating pulse and the short MW pulse (see Fig. 1f). The typical ODMR spectra shown in Fig. 1g shows the ZFS shift for various waiting times $t_w$. The temperature as a function of the waiting time in Fig. 1h is again well fitted by the same rate equations as used for fitting Fig. 1e, with the cooling time about 2 µs (the same as the heating time, as determined by the heat diffusion rate of the carbon film).

The scheme of HiT-ODMR is shown in Fig. 2a. We use a weak 532 nm green laser pulse to initialize the NV centers to the $|0\rangle$ state at room temperature, then apply a strong NIR laser pulse for rapid heating, then the MW pulse with a duration of 30 ns is applied at the end of the 5 µs NIR heating pulse so that the spin manipulation is carried out at the stationary temperature, and finally carry out the readout by another weak 532-nm laser pulse after the ND has cooled down. The MW pulse is much shorter than the heating/cooling time so that the temperature variation during the spin manipulation is negligible. The saturated temperature is tuned by varying the heating laser power. Figure 2b shows the ODMR spectra of an ND for various heating laser powers. The measurement is under zero magnetic field. The ZFS $D$ shifts to lower frequencies with increasing the heating power. The contrast of ODMR persists for $D$ down to $2758 \pm 1$ MHz.

To determine the temperature for different heating laser powers, we use the $D(T)$ curve measured in ref. [15] for $T < 700$ K (corresponding to $D > 2815$ MHz). For higher temperatures (i.e., smaller $D$), we extract the temperature by extrapolation of the exponential cooling after the heating pulse (see Methods and Supplementary Note 6 for details). The temperature immediately after the heating pulse (the peak temperature) is determined by extrapolation of the exponential cooling using the temperatures at different waiting times $t_w$ (which are calibrated by $D(T)$ as they are below 700 K). Figure 2c shows the temperature dependence of the ZFS $D(T)$ obtained by the extrapolation approach for several NDs, in comparison with the result from ref. [15]. The highest temperature at which the ODMR signal is observed (corresponding to the lowest $D$, $2758 \pm 1$ MHz) is $1004 \pm 24$ K.

**Spin relaxation at high temperature.** Figure 3a shows the pulse sequence for measuring the spin relaxation times of NV centers in NDs at high temperatures (when the NIR laser is kept on). Each measurement cycle contains two units—in the first unit the spins are initialized to the $m_s = 0$ state by a 532-nm laser pulse and then heated to a high temperature by an NIR laser pulse (duration = 5 µs + $t_r$), after a fixed cooling time $t_c = 3$ µs, another 532-nm laser pulse is used for spin readout. The second unit is the same as the first one except that a MW $\pi$ pulse is applied $t_r$ before the end of the NIR laser pulse ($t_r$ is shorter than the NIR laser pulse duration and the temperature has reached the saturated value when the MW pulse is applied). The time for measuring the spin relaxation at high temperature (when the NIR laser is kept on), $t_r$, is the time between the MW $\pi$ pulse and the end of the NIR pulse (see Fig. 3a). The photon counts from the two units, $P_0(t_r)$ for the first and $P_{-1}(t_r)$ for the second (see Supplementary Fig. 9), are recorded as functions of $t_r$. Their normalized difference $\Delta P(t_r) \equiv \frac{P_0(t_r) - P_{-1}(t_r)}{P_0(t_r) + P_{-1}(t_r)}$ is used to single out spin signals from overall photon count variations (due to, e.g., charge dynamics of NV centers). Figure 3b plots $\Delta P(t_r)$ for various heating powers, showing the spin relaxation for various temperatures (summarized in Fig. 3c).

At high temperature, the spin relaxation time $T_1$ (Fig. 3c) is well fitted by $T_1^{-1} = AT^n$ with $A = 8 \times 10^{-12}$ s$^{-1}$ K$^{-n}$ and $n = 5.6 \pm 0.5$, which is close to the temperature dependence due to the two-phonon Raman processes ($\propto T^5$)[28]. The $T^5$ temperature dependence of the longitudinal relaxation rate is determined by the dimension of the NV system and symmetry of the diamond lattice[28]. This model has been previously verified in diamond at temperatures up to 475 K[22]. Our experimental results show that this model is still valid up to near 1000 K. The spin relaxation time saturates at ~100 µs at room temperature, which we ascribe to the electric and magnetic noises in the NDs (Supplementary Note 7).

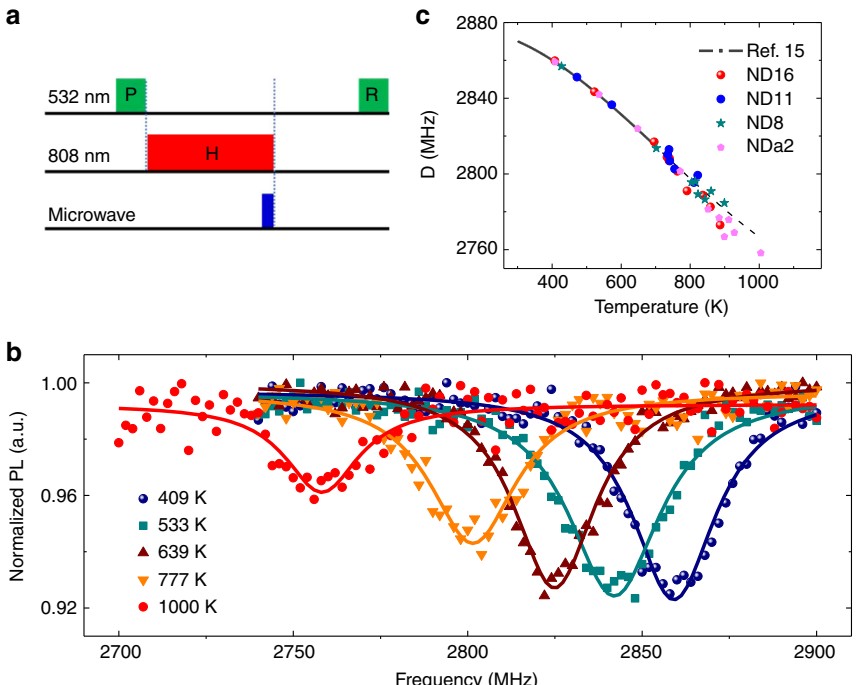

**Fig. 2** High-temperature optically detected magnetic resonance (HiT-ODMR) of nitrogen-vacancy (NV) centers in nanodiamonds (NDs). **a** Pulse sequence for HiT-ODMR, with the same notations as in Fig. 1c. The microwave pulse is applied at the end of the NIR laser pulse. **b** ODMR spectra of an ND for various heating laser powers (corresponding temperatures indicated, see Methods and Supplementary Note 6 for details). The NIR laser pulse duration $\tau_h$ is fixed to be 5 μs. The stationary temperature is proportional to the NIR laser power. **c** The temperature dependence of the zero-field splitting (ZFS) $D(T)$ obtained by extrapolation of the exponential cooling dynamics for four different NDs (indicated by different symbols), in comparison with the results from ref. [15] (solid line for $T < 700$ K). The dashed line is the fitting formula in ref. [15] extended to 1000 K

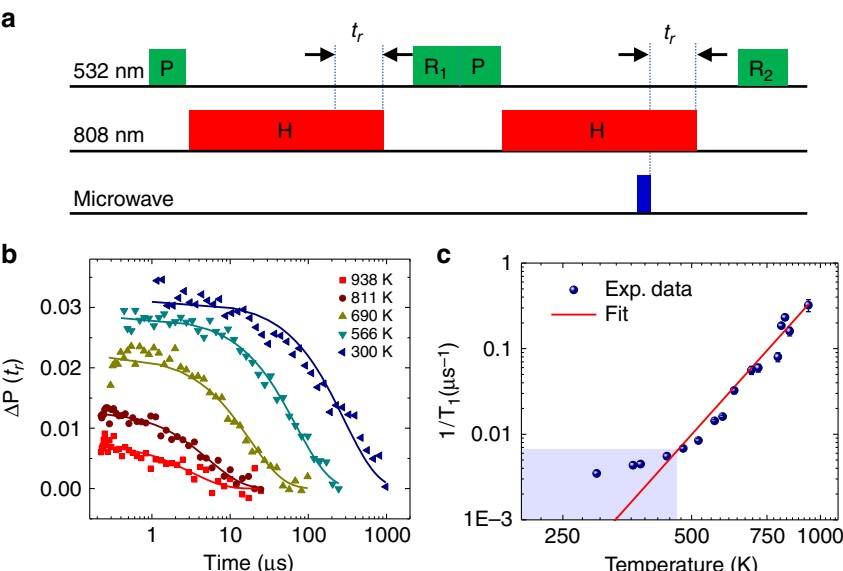

**Fig. 3** Spin relaxation of nitrogen-vacancy (NV) centers in a nanodiamond (ND) at high temperatures. **a** Pulse sequence. After spin polarization by the short green laser pulse at low temperature, the ND is heated to a high stationary temperature by an NIR pulse (5 μs). The ND is maintained at that temperature for $t_r$ with the NIR laser kept on, and then after a fixed period of cooling (3 μs) the NV center spins are read out. To eliminate the fluorescence changes that are not related to spin states, the fluorescence difference between the $m_s = 0$ and $m_s = -1$ states are taken as the signal, with a microwave π pulse applied in the latter case. **b** Spin-dependent fluorescence as a function of relaxation time, measured for the NV center spins of an ND (NDa2 in Fig. 2c) at different temperatures. **c** Spin relaxation rate ($1/T_1$) as a function of temperature. The solid blue circles are experimental data. The red line is the power-law fitting ($T_1^{-1} = AT^n$) in the high temperature regime with $A = 8 \times 10^{-12}$ s$^{-1}$ K$^{-n}$ and $n = 5.6 \pm 0.5$. At lower temperature (shadowed region), the relaxation rate saturates. Error bars corresponding to the fitting errors

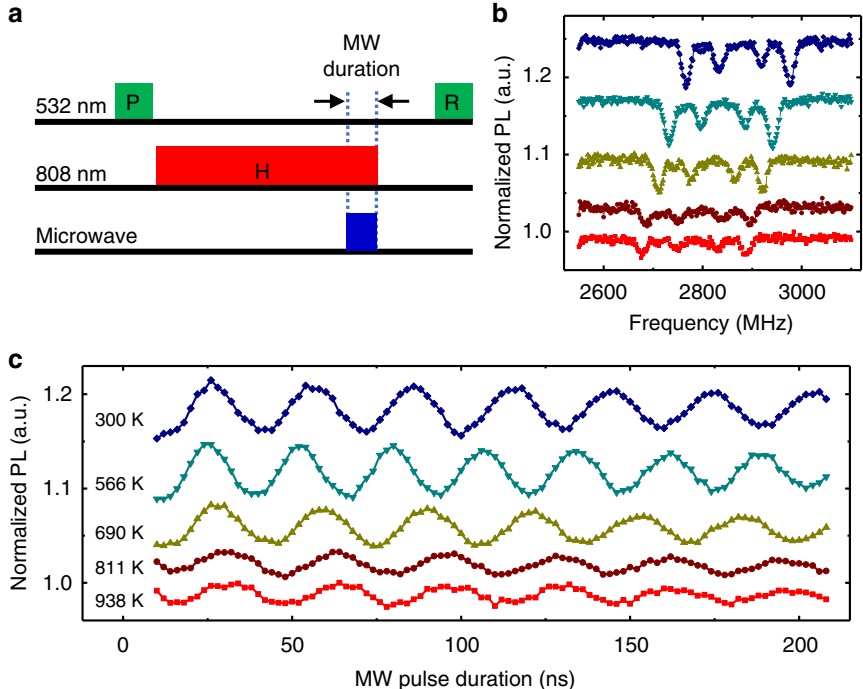

**Fig. 4** Rabi oscillations of nitrogen-vacancy (NV) center spins in a nanodiamond (ND) at temperatures approaching to 1000 K. **a** Pulse sequence. The microwave driving is applied after the temperature reaches the stationary value (the NIR laser pulse duration being 5 µs). **b** Optically detected magnetic resonance (ODMR) spectra and **c** Rabi oscillations of NV center spins in the ND at different temperatures (indicated by different colors as shown in **c**). An external magnetic field of about 38 Gauss is applied in these measurements. The temperature is extracted from the zero-field splitting (ZFS) $D$ from the $D$-$T$ relation in Fig. 2c

**Rabi oscillation at high temperature**. Rabi oscillation is the basis for different kinds of quantum control of spins (such as quantum gates and dynamical decoupling). Figure 4a shows the pulse sequence for observing the Rabi oscillation of NV center spins in NDs at high temperatures. After the spin initialization by a weak 532-nm laser pulse, the ND is heated up by a strong NIR laser pulse for 5 µs and kept at a high temperature during the Rabi oscillation (the MW driving is applied after the temperature is saturated and with the NIR laser pulse kept on). The 532 nm laser pulses are applied at lower temperatures for spin polarization and readout. A weak magnetic field (38 Gauss) is employed to split the resonances of the NV centers belong to different crystallographic orientations (Fig. 4b). The Rabi oscillation is observed with a MW pulse set resonant with the first transition (the one of the lowest frequency in Fig. 4b, for its relatively large contrast). Though its visibility decreases with increasing the temperature, the Rabi oscillation persists at temperature up to 938 K with no significant change of the coherence time (Fig. 4c). The visibility loss at higher temperature may be caused by the spin relaxation during the temperature rising (the duration of the MW pulse is much shorter than the heating duration).

**High temperature nano-magnetometry**. A potential application of HiT-ODMR of NDs is to study the magnetism of single nanoparticles, where interesting phenomena such as magnetic phase transitions and superparamagnetic blocking often occur at temperatures above 500 K, which is beyond the working temperature range of most existing nano-sensors. We demonstrate the high temperature nano-magnetometry by measuring the magnetic properties of a single nickel nanoparticle (Ni NP)[29]. For the proof-of-the-principle experiments, this material system is chosen for its stability and reproducibility to avoid potential complications in data analysis. As shown in the inset of Fig. 5b,

an ND located near a Ni NP is used as the sensor. The stray magnetic field from the Ni NP causes splitting and broadening of the NV center spin resonances (Fig. 5a). The broadening is due to the gradient of the stray field in the range of the ND, which contains about 500 randomly distributed NV centers[29]. With increasing the temperature, the ODMR spectra become narrowed, indicating the demagnetization of the Ni NP.

The ODMR splitting (Fig. 5b) due to the Ni NP is measured in four rounds of cooling processes from a high temperature $T_H$ (which is chosen to be above or below $T_C$) to a low one $T_L$ (which is chosen to be below $T_C$). A magnetic phase transition at $T_c = 615 \pm 4$ K is clearly seen. The magnetization in a cooling process is nearly the same as the previous round if the cooling starts at $T_H < T_C$ (Rounds 2 and 4 in Fig. 5b). When the Ni NP is cooled down from above $T_C$ (Rounds 1 and 3 in Fig. 5b, in which $T_H > T_C > T_L$), the resultant magnetization at low temperature is different in different cooling rounds due to the spontaneous symmetry breaking in the phase transition.

**Discussion**

HiT-ODMR opens up several opportunities for nano-thermometry and nano-magnetometry. Compared with other quantum sensors[1] like SQUIDs, trapped ions, atomic vapors or scanning probes, NV spins in diamond combine the merits of good sensitivity, nanoscale spatial resolution, and a wide range of working conditions[30] (including cryogenic temperature to about 1000 K, from vacuum to ambient conditions, and high pressure[31]). Using the photon counts (8 Mps), the ODMR width (10 MHz, mainly due to the internal local charge and strain-induced broadening[32]), and the ODMR contrast (5%) of the NDs we have measured at 1000 K, we estimate the temperature sensitivity to be about 250 mK Hz$^{-1/2}$ based on the ZFS shift $dD/dT \approx 240$ kHz K$^{-1}$ (determined from data in Fig. 2c) and the magnetic field

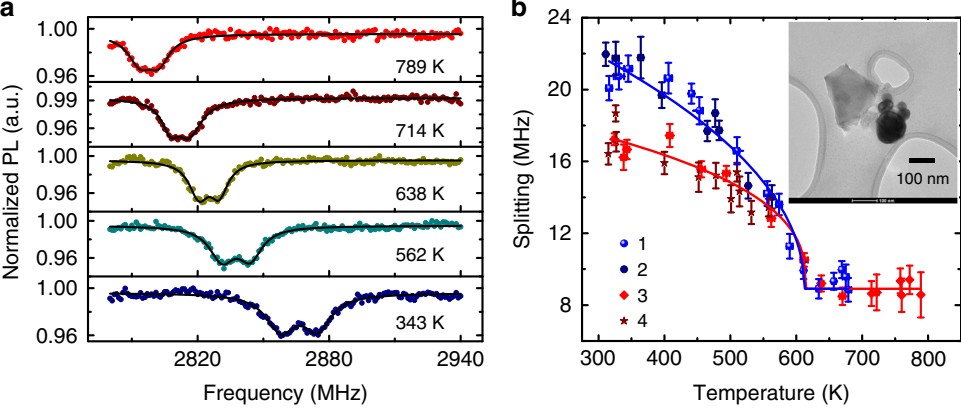

**Fig. 5** Magnetic phase transition of a single nickel nanoparticle. **a** Typical zero-field optically detected magnetic resonance (ODMR) spectra of a nanodiamond (ND) located close to a nickel nanoparticle (Ni NP) at various temperatures. The splitting and extra broadening are induced by the magnetization of the nearby Ni NP. The pulse sequence is the same as in Fig. 2a. **b** Spontaneous magnetization of the Ni NP in several rounds of cooling, the measurement order is Round 1 ($T_H$ = 669 K to $T_L$ = 316 K) → Round 2 (563 K to 311 K) → Round 3 (789 K to 324 K) → Round 4 (558 K to 326 K). The solid lines are fitting curves with a function of $\sim (1 - T/T_C)^\beta$ to extract $T_C$ (615 ± 4 K) of this Ni NP. Inset: Transmission electron microscopy (TEM) image of the measured ND (light gray) and the nearby Ni NP (black sphere). Error bars corresponding to the standard fitting errors

sensitivity to be 2.5 µT Hz$^{-1/2}$ (see Methods and Supplementary Note 10). The temperature sensitivity can be further improved by adopting advanced sensing protocols like D-Ramsey[18–20] and magnetic criticality enhanced thermometer[29] and by using NV centers of good spin coherence and fast temperature control.

The current scheme is limited by the speed of laser heating and diffusion cooling as compared with the relaxation time of NV center electron spins at high temperature. To enable the fast heating and cooling, the applications are restricted to samples of small sizes (e.g., microparticles). However, such sample sizes do not impose a severe constraint on many applications in materials sciences and engineering studies[33] or in device physics. The temperature range of the spin coherence control in diamond can be pushed to even higher. In current experiment, the spin relaxation time (~5 µs, Fig. 3c) at about 1000 K is comparable to the laser heating time (~2 µs, Figs. 1e, h) in our setup, making it challenging to observe the ODMR above 1000 K. This limit can be overcome by either increasing the heating and cooling speed or by increasing the spin relaxation time. The latter can be realized, e.g., by resorting to diamond samples with better coherence (such as nano-pillars with single NV centers).

## Methods

**Samples**. NDs with ensemble NV centers (average size ~140 nm) are purchased from Adámas. Each ND contains about 500 NV centers. NDs dispersed on a TEM grid (Ted Pella) are attached to the amorphous carbon film. NDs are stable on the carbon film, with no change observed during the whole measurement. Ni NPs are obtained by ball milling of Ni powder (3–7 µm, Strem Chemicals). High temperature annealing (973 K, 2 h) is performed in 10% $H_2$/Ar after ball milling to improve the crystallinity of the NPs. After size sorting by centrifugation, Ni NPs with size of about 100 nm are dispersed on a TEM grid. The composition of single Ni NPs is verified by energy-dispersive X-ray spectroscopy (EDX). The locations of Ni NPs and NDs in the proximity are identified by TEM imaging using the unique net patterns of the amorphous carbon film (Supplementary Fig. 1). In confocal fluorescence imaging, the NDs are much brighter than the carbon film and the Ni NPs. The overlap between the TEM and confocal images is employed to locate NDs with Ni NPs in the proximity (Supplementary Fig. 2).

**Sample holder**. To avoid oxidation of the amorphous carbon film during laser heating, the sample is protected in an Ar atmosphere in the high temperature measurements (see Supplementary Fig. 3). The chamber is built on a confocal dish, with a reusable cover. The bottom of the dish is removed, then glued to the printed circuit board with MW transmission lines. The TEM grid is fixed in the chamber, and then the opened chamber is placed in glovebox of an Ar atmosphere for about

10 h to be filled with Ar. Finally, the chamber is closed with the dish cover and sealed with additional glue before being taken out from the glovebox for ODMR measurement.

**ODMR setup**. The optical system contains (1) spin polarization with a green laser (532 nm), (2) spin readout with fluorescence collection under 532 nm laser excitation, and (3) local and instantaneous heating with an NIR laser (808 nm), as shown in Supplementary Fig. 4. The 532-nm laser is modulated by an acousto-optic modulator (AOM) and coupled into a microscopy frame through a single-mode fiber. A pair of galvo mirrors control the focusing position (X-Y) of the green laser, and a piezo stage controls the focus depth of the oil objective (NA = 1.35). The fluorescence of NV centers is collected by the same objective and passed through two dichroic mirrors (one for the green laser and another for the NIR) and filters, then detected by a single photon counting module (Excelitas). The NIR laser is independently controlled with an AOM and a pair of galvo mirrors, adjusted to overlap with the green laser. A pair of moving lenses are used to compensate the chromatic aberration between the two lasers. The amplitude and frequency of MW pulses are controlled by the signal generator (N5181A, Agilent) and their shapes are modulated by an RF switch (ZASW-2-50DR, Mini Circuits). The laser excitation, NIR heating, MW manipulation, and fluorescence readout are synchronized with TTL signals from a pulse generator (PulseBlasterESR-PRO, SpinCore).

**Temperature calibration**. The temperature below 700 K is calibrated with the D-T relation from ref. [15]. For temperatures higher than 700 K, we measure the ZFS D for different waiting times $t_w$ (see Fig. 1f) and extract the temperature for the ZFS using the D-T relation in ref. [15] for D > 2815 MHz (i.e., T < 700 K). Then, the temperature right after the heating pulse is deduced by extrapolation with the assumption of exponential cooling (which is verified in experiments). The ODMR spectra at different $t_w$ are measured in a cycling manner (sequence in Supplementary Fig. 7), which is designed to remove possible effects due to aging of carbon films (which would reduce the heating efficiency), drifting of NIR laser focus spot, and laser power fluctuations (see Supplementary Note 6 for details). We have verified the method by comparing the temperature obtained by extrapolation with that determined by the D-T relation for T < 700 K (Fig. 2c). The examples of temperature calibration are shown in Fig. 6 and the results are summarized in Supplementary Table 1.

In the spin relaxation and Rabi oscillation experiments (Figs 3 and 4, respectively), the D value is determined by averaging the frequencies of the two outmost resonances and then the D-T curve in Fig. 2c is used to determine the temperature.

## Data availability

Data supporting the findings of this study are available within the article and its Supplementary Information and from the corresponding authors upon request.

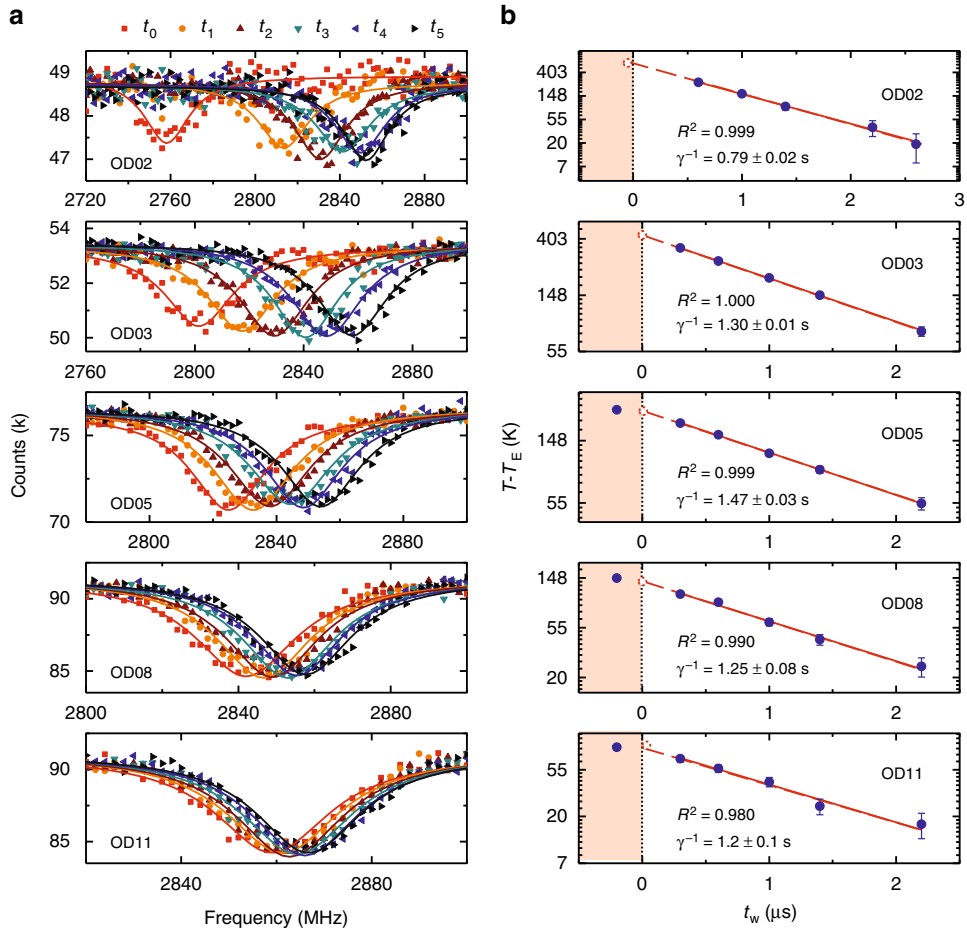

**Fig. 6** Temperature calibration by extrapolation of exponential cooling. **a** Optically detected magnetic resonance (ODMR) spectra for various waiting times $t_w$ between the NIR laser pulse and the microwave pulse (indicated by different symbols), $t_w = \{-0.2, 0.6, 1.0, 1.4, 2.2, 2.6\}$ μs for OD02, and $t_w = \{-0.2, 0.3, 0.6, 1.0, 1.4, 2.2\}$ μs for the others. The pulse sequence in Supplementary Fig. 7 is used. No external magnetic field is applied. The NIR laser power is increased from bottom to top, which is tuned by an optical attenuator (OD0 means full power). The ODMR spectra are fitted with a Lorentzian dip to extract the zero-field splitting (ZFS) $D$. **b** The temperature of the nanodiamond (ND) (relative to the environment temperature $T_E$) as a function of the waiting duration $t_w$ for various NIR laser powers. The temperature lower than 700 K (solid symbols) is obtained using the $D$-$T$ relation in ref. [15]. All the data for $t_w > 0$ are well fitted with an exponential decay function. By extrapolation, the temperature immediately after the heating pulse is obtained (red open circles). The plateau before the end of the heating pulse (marked by the vertical dotted line) is due to the AOM delay ($\approx 200$ ns). Error bars corresponding to the standard fitting errors

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

## Acknowledgements

This work was supported by the Hong Kong Research Grants Council-Collaborative Research Fund Project C4006-17G, Hong Kong Research Grants Council—General Research Fund Project 14319016, and The Chinese University of Hong Kong Group Research Scheme 2017–2018 Project 3110126.

## Author contributions

R.-B.L. conceived the idea. R.-B.L. and Q.L. supervised the project. R.-B.L., Q.L. and G.-Q.L. designed the experiment. G.-Q.L. noticed the laser heating effect of carbon thin films. G.-Q.L. and N.W. set up the HiT-ODMR system and carried out the ODMR experiments. G.-Q.L. prepared the NDs and the Ar-filled sample chambers. X.F. characterized NDs on carbon thin films, and prepared and characterized Ni NPs, and NDs and Ni NPs on carbon films. G.-Q.L., R.-B.L. and Q.L. analyzed the data. G.-Q.L., R.-B.L. and Q.L. wrote the paper. All authors commented on the manuscript.

## Additional information

**Competing interests:** The authors declare no competing interests.

