## [Peer Review File · Nature Communications]

Editorial Note: This manuscript has been previously reviewed at another journal that is not operating a transparent peer review scheme. This document only contains reviewer comments and rebuttal letters for versions considered at Nature Communications. Mentions of the other journal have been redacted.

REVIEWERS' COMMENTS:

Reviewer #1 (Remarks to the Author):

Having read through the authors detailed responses I am satisfied with the manuscript in its current form and recommend publication in Nature Communications as is.

Reviewer #2 (Remarks to the Author):

The authors clearly replied to my concerns following their first submission to [redacted]. The results look valid, interesting, and I would even say that the method is elegant. This article is worth publishing.

However, the fact that the measurement are indirect (extrapolation at high temperature of a measurement carried out at low temperature assuming an exponential decay), limited to small micro- nanoscale systems, and limited to short heating time scale (a few microseconds) make this study not as useful as it seems to be upon reading the title, and applicable to a very small amount studies. If this article could be well cited because of a nice apparent achievement, I doubt this technique will be reproduced by the community to make science advance.

This being said, whether this article is suited for a journal such as Nature Communication may be up to the editorial board to decide.